# Dose-Response Relationships between Breastfeeding and Postpartum Weight Retention Differ by Pre-Pregnancy Body-Mass Index in Taiwanese Women

**DOI:** 10.3390/nu12041065

**Published:** 2020-04-11

**Authors:** Alexander Waits, Chao-Yu Guo, Yan-Shing Chang, Li-Yin Chien

**Affiliations:** 1Institute of Public Health, National Yang-Ming University, Taipei 112, Taiwan; alexwaits@ym.edu.tw (A.W.); cyguo@ym.edu.tw (C.-Y.G.); 2Tao Yuan General Hospital, Ministry of Health and Welfare, Taoyuan 320, Taiwan; 3Child and Family Health Department, Florence Nightingale Faculty of Nursing, Midwifery and Palliative Care, King’s College, London WC2R 2LS, UK; yan-shing.chang@kcl.ac.uk; 4Institute of Community Health Care, National Yang-Ming University, Taipei 112, Taiwan

**Keywords:** exclusive breastfeeding, pre-pregnancy body mass index, Taiwan, obesity

## Abstract

Postpartum weight retention (PWR) is a risk factor for future obesity. The role of breastfeeding in reducing PWR is not fully understood. We examined the relationship between PWR and the duration of exclusive/partial breastfeeding in 52,367 postpartum women from 2012–2016 Taiwan national breastfeeding surveys. The women were interviewed at 7–14 months postpartum. Non-linear models were fit to examine the association between PWR and breastfeeding duration. PWR adjusted means and 95% confidence intervals were plotted and compared for the duration of exclusive/partial breastfeeding in the total sample and between pre-pregnancy body-mass index (BMI) groups (underweight, normal, overweight, and obese). Women who breastfed exclusively for >30 days showed significantly lower PWR than those who did not breastfeed and those who breastfed partially for the same duration, thereafter each additional duration of 30 days being associated with an average of 0.1–0.2 kg less PWR. Women who breastfed partially for 120 days showed lower PWR than those who did not or those who ceased to breastfeed, thereafter each additional duration of 30 days being associated with an average of 0.1 kg less PWR. Duration of breastfeeding needed to achieve significantly less PWR differed between pre-pregnancy BMI groups, but the effect of exclusive breastfeeding appeared earlier in the normal weight group. Women with obesity who breastfed exclusively for >30 or partially for >180 days, had lower PWR than non-obese groups. The observed dose–response relationship between breastfeeding duration and PWR supports the “every feeding matters” approach in breastfeeding promotion. The larger effect of exclusive and partial breastfeeding on PWR in women with obesity may draw special attention of breastfeeding promotion.

## 1. Introduction

Many postpartum women keep some of the weight gained during gestation, which prevents them from returning to their pre-pregnancy weight [1,2,3,4]. This retained weight, known as postpartum weight retention (PWR), is expressed as the difference between pre- and post-pregnancy weight [5]. PWR has been associated with an increased risk of developing obesity within one year postpartum [6]. Moreover, PWR showed a cumulative effect for weight gain in subsequent pregnancies, amplifying the risks for lifelong maternal obesity [1]. Thus, PWR may also contribute to a higher risk of obesity in the offspring [7], creating a “vicious cycle” of obesity [2]. Therefore, within the context of globally growing rates of obesity [8], PWR poses an important concern for public health. The United States Institute of Medicine recommends that women return to their pre-pregnancy weight within 6–12 months of delivery [8], however, only a few women reach this goal [1]. In a meta-analysis of 17 studies, average PWR at six months postpartum was 3.5 ± 6.2 kg, with 73% of women retaining some weight [3].

Higher PWR has consistently been associated with higher gestational weight gain (GWG), which could explain 30%–35% of variation in PWR after six months postpartum [9,10,11]. A lower pre-pregnancy body-mass index (BMI) was associated with a higher PWR [3,10], while obese and overweight mothers were less likely to return to their pre-pregnancy BMI regardless of GWG [11]. This implies that insufficient or excessive BMI prior to conception is likely to be associated with higher PWR. Additional risk factors included maternal age, higher parity and non-Asian ethnicity [9], dietary intake and longer television viewing [12], and a lack of physical activity and social engagement [13].

While benefits of breastfeeding for maternal health have been well established [14], its contribution to PWR remains ambiguous [15]. A critical evaluation of 45 prospective and retrospective studies in developed countries provided inconclusive evidence to associate breastfeeding with PWR. Breastfeeding for <3 months had a minimal effect on PWR. Only 7 of 15 studies that examined breastfeeding for ≥6 months postpartum reported consistent weight loss, yet studies with rigorous methodology found small, but significant negative associations between breastfeeding duration/frequency and PWR [15]. More recent cohort studies demonstrated that breastfeeding for >3 [13] and 6 months [10] was associated with lower PWR at 6 months postpartum. Assessments of dose–response relationships between breastfeeding and PWR were inconclusive, and non-linear relationships were suggested, since higher breastfeeding frequency was associated with less weight loss from 3 to 6, but with greater weight loss from 9 to 12, months [16,17,18,19].

Some studies in Asian countries, including Taiwan, focused on PWR, however only a few examined its relationship with breastfeeding. A review of 12 Asian studies found that the average PWR at six months ranged from 1.6 kg to 4.1 kg, with Chinese, Taiwanese, and Korean women retaining more postpartum weight than women in other Asian countries [20]. To the best of our knowledge, only two Taiwanese studies explored the relationship between breastfeeding and PWR. A cohort study with 120 women did not find an association between breastfeeding and PWR [21], however a more recent study with a larger sample size (*N* = 461) showed a significant negative association between exclusive breastfeeding and PWR [22].

The role of breastfeeding in PWR remains unclear, and only limited research in Asian populations has addressed this issue. Herein, we used a nationally representative sample of Taiwanese women to assess the association of breastfeeding with PWR after six months postpartum. We hypothesized a possible dose–response relationship between breastfeeding duration and PWR. As previous studies reported the risks of higher PWR for underweight and overweight/obese women [3,10,11], we also examined whether the relationship differed by type of breastfeeding and by pre-pregnancy weight status. Our findings are expected to enhance the understanding of the relationship between breastfeeding and PWR and contribute to breastfeeding promotions and maternal weight management.

## 2. Materials and Methods

### 2.1. Study Design and Participants

The data were retrieved from five cross-sectional national breastfeeding surveys, conducted annually by Taiwan Health Promotion Administration from 2012 to 2016. The original surveys were approved by the joint institutional review board. Our study used anonymous data without linkable ID. Telephone interviews with structured questionnaires were administered at 7–14 months postpartum to randomly drawn women >20 years of age who gave birth to a live baby and the baby was alive at the time of interview. The questionnaires were reviewed by a committee of experts, and validity was tested with a pilot study. Questions included information on breastfeeding type and duration, obstetric and neonatal outcomes related to the index pregnancy, and maternal sociodemographic factors. A disproportionate probability sampling method, based on annual number of births in 25 Taiwan counties, was applied so that county-specific breastfeeding rates could be estimated for counties with a small number of births. Sample sizes ranged from 12,071 to 12,553 in each year with participation rates of 35.3–37.7% and sampling errors of <1%.

### 2.2. Measurements

Breastfeeding duration and exclusivity were assessed through questions: (1) “Till what time (days postpartum) did you breastfeed?” (2) “When did you start adding formula, solid foods, or other liquids to feed the baby?” (3) “When did you completely stop breastfeeding?” If the answer was “I am still breastfeeding”, breastfeeding duration was taken as days postpartum at the time of the interview. Exclusive breastfeeding was defined as whether infants were fed with human milk only, with no use of formula, solids, or other liquids. Partial breastfeeding was defined as whether infants received any formula, solids, or other liquids in addition to human milk. Three continuous variables were calculated to describe duration of exclusive breastfeeding, duration of partial breastfeeding, and time postpartum of ceasing breastfeeding completely.

Survey respondents were asked to report their pre-pregnancy weight, weight at delivery, and current weight and height at the time of the interview. PWR was calculated by subtracting pre-pregnancy weight from the weight reported at the time of the interview. Since interviews were held at different times postpartum, ranging from 7 to 14 months, we compared PWR means for each month with an analysis of variance (ANOVA), resulting in a non-significant difference (*p* = 0.161) (Table 1). Pre-pregnancy BMI in kg/m² was calculated from self-reported pre-pregnancy weight and height, which was found as a reliable and valid method for population-based research in women of reproductive age [23]. It has been advised by World Health Organization experts to define local categorization of BMI for Asian populations, accounting for different body mass composition from Westerners, and adjusting for the difference in and diversity of populations in different Asian regions [24]. We therefore employed BMI cutoff points used in Taiwan: 18.5, 24, and 27 kg/m^2^ for underweight, normal, overweight, and obese, respectively (Taiwan Health Promotion Administration, 2018).

### 2.3. Data Analysis

We calculated means and standard deviations for continuous variables. We compared PWR means between the categories within each variable with a two-sample t-test for dichotomous and an ANOVA for categorical variables with more than two groups. Significance level was set to two-sided α = 0.05.

Generalized linear models were fitted with identity link function and robust variance estimates to assess the association of PWR with the duration of exclusive and partial breastfeeding. We adjusted the models for maternal age, education, employment, country of origin, gestational weight gain, parity, multiple gestation, cesarean section, and the month postpartum, when the interview was conducted; newborn sex, year of birth, weight and health status at birth. A quadratic term for breastfeeding duration was introduced in the models to account for non-linear relationships and improve model fit. The adjusted means of PWR in kilograms and their 95% confidence intervals (CI) were plotted for breastfeeding duration in days. Non-linearity of our models allowed assessment and comparison of slopes only at specific values of the independent variables, therefore, we compared PWR means for every 30 days of breastfeeding to assess statistical significance. The series of 95% CI in consecutive time points could indicate significant dose–response relationships when confidence intervals did not overlap. Comparison between the curves was based on the 95% CI at each time point. Overlapped confidence intervals suggested no significant difference between the groups represented by the curves. Models were stratified by four categories of pre-pregnancy BMI to compare between partial and exclusive breastfeeding for up to 180 days in each BMI group. An additional comparison was performed between BMI groups on the relationships of PWR with exclusive breastfeeding up to 180 days and with partial breastfeeding up to 360 days.

All models were weighted to enhance the representation of the sampled population while accounting for disproportionate probability sampling methods. Residual plots for all the models are available in the Appendix A. Although multiple gestation can influence GWG [8] and affect breastfeeding behavior [14], we expected its negligible effect on PWR, especially given its small proportion (2.6%) in our dataset. The adjustment for multiple gestation and gestational weight gain in our models provided similar results to the sensitivity analysis with singletons only (Appendix A). Since our conclusions remained unaffected, we present the results based on the full dataset. All the analyses were performed with Stata Statistical Software: Release 15 (StataCorp. 2017. StataCorp LLC: College Station, TX, USA).

## 3. Results

### 3.1. Participants Characteristics

We analyzed 52,367 respondents, most of whom were native Taiwanese (94.9%) with singletons (97.4%) of term gestation (91.9%). The largest age groups comprised 30–34 years (42.9%) and ≥35 years (27.9%). Over half of the respondents had been educated at university level or higher (55.9%), worked outside home (57.9%), and had a vaginal delivery (64.7%). The majority of respondents had a normal pre-pregnancy BMI (66.7%), while 17.0% were underweight, 10.4% were overweight, and 5.9% were obese. Women of older age and higher education tended to retain less weight postpartum (Table 1). Mean duration of breastfeeding was 82 ± 75 days for exclusive breastfeeding and 177 ± 112 days for partial breastfeeding. Mean PWR was 2.40 ± 4.15 kg. The obese group retained significantly (*p* < 0.001) less weight and breastfed for a significantly (*p* < 0.001) shorter duration than any other BMI group (Table 2). Only 36.6% of women in our study returned to their pre-pregnancy weight at 7–14 months postpartum.

### 3.2. Duration and Exclusivity of Breastfeeding

Significant dose–response relationships were observed between PWR and exclusive breastfeeding at each consecutive interval of 30 days starting from 30 days postpartum (Figure 1). For example, those who exclusively breastfed for 30 days showed a significantly lower PWR mean of 2.69 kg (95% CI = 2.64–2.74) than those who had zero days of exclusive breastfeeding (2.91 kg, 95% CI = 2.85–2.98), and so on for 60 versus 30, 90 versus 60, 120 versus 90, and 150 versus 120 days of breastfeeding. Women who exclusively breastfed for 180 days had a marginally significant lower PWR mean of 1.84 kg (95% CI = 1.77–1.90) compared to women who did so for 150 days (1.98 kg, 95% CI = 1.92–2.03). PWR means lowered significantly for those women who partially breastfed for 120 days (2.76 kg, 95% CI = 2.70–2.81) and 180 days (2.55 kg, 95% CI = 2.49–2.60) postpartum compared to those who ceased breastfeeding within 30 days postpartum (2.91 kg, 95% CI = 2.81–3.02). Women who exclusively breastfed for at least 30 days retained significantly less weight than those who breastfed partially during the same period or ceased breastfeeding earlier. Those who breasted partially for at least 120 days retained less weight than those who ceased breastfeeding earlier (Figure 1). All PWR means and 95% CI are presented in the Appendix A.

### 3.3. Comparison of Exclusive and Partial Breastfeeding by Pre-Pregnancy BMI

Figure 2 compares exclusive and partial breastfeeding within each BMI group. In the normal pre-pregnancy BMI group, exclusive breastfeeding was associated with significantly lower PWR compared to partial breastfeeding at any time point of breastfeeding duration, starting from 30 days postpartum (Figure 2B). Similar relationships between partial and exclusive breastfeeding were observed for underweight (Figure 2A) and overweight (Figure 2C) groups starting from 60 days, and for obese (Figure 2D) starting from 90 days postpartum. An overlapping of 95% CI suggested no significant differences between partial-breastfeeding and non-breastfeeding groups in the three BMI groups, except the normal BMI group, where partial breastfeeding for 150 and 180 days showed significantly lower PWR (Figure 2B). The lines for partial- and non-breastfeeding groups in the obese group (Figure 2D) overlapped and peaked at 90 days postpartum. All PWR means and 95% CI are presented in the supplementary file.

Figure 3A compares exclusive breastfeeding by BMI groups. A significant dose–response relationship between PWR and exclusive breastfeeding at each consecutive interval of 30 days starting from 30 days postpartum was observed in mothers with a normal pre-pregnancy BMI, and in the three other BMI groups, where PWR means decreased significantly for those who exclusively breastfed for 60 and 120 days postpartum. Within each pre-pregnancy BMI group, the PWR mean of mothers who breastfed exclusively for 180 days was significantly lower than that of mothers with zero days of exclusive breastfeeding: 1.90 kg (95% CI = 1.82–1.98) versus 2.96 kg (95% CI = 2.89–3.04) for normal BMI; 2.03 kg (95% CI = 1.91–2.16) versus 2.75 kg (95% CI = 2.62–2.89) for underweight; 1.65 kg (95% CI = 1.41–1.88) versus 3.04 kg (95% CI = 2.82–3.27) for overweight; and 0.64 kg (95% CI = 0.31–0.98) versus 2.61 kg (95% CI = 2.31–2.92) for obese. Relationships between exclusive breastfeeding and PWR at each time point postpartum were not significantly different between the three BMI groups, except for the obese one. Women with obesity who exclusively breastfeed for longer than 30 days were likely to have significantly lower PWR than any other non-obese group. All PWR means and 95% CI are presented in the Appendix A.

Figure 3B presents a significant dose–response relationship between PWR and partial breastfeeding in mothers with normal pre-pregnancy BMI at each consecutive interval of 30 days, starting from 150 days postpartum. PWR means also lowered significantly at 210, 270, and 330 days for underweight, and at 150, 240 and 300 days for overweight. The obese group presented a peak at 90 days, and significantly lower PWR at 270 and 330 days postpartum. There was no significant difference in PWR means between underweight, normal, and overweight groups, while women with obesity who exclusively breastfeed for >180 days were likely to have significantly lower PWR than normal or underweight groups, and if for >240 days, lower than the overweight group, respectively. Within each pre-pregnancy BMI group, PWR means of mothers who breastfed partially for 360 days were significantly lower than those of mothers who did not breastfeed at all: 1.42 kg (95% CI = 1.29–1.54) versus 2.98 kg (95% CI = 2.85–3.11) for normal BMI; 1.71 kg (95% CI = 1.51–1.91) versus 2.73 kg (95% CI = 2.53–2.94) for underweight; 1.15 kg (95% CI = 0.76–1.55) versus 3.28 kg (95% CI = 2.92–3.64) for overweight; and −0.20 kg (95% CI = −0.74–0.34) versus 2.3 kg (95% CI = 1.8–2.7) for obese. All PWR means and 95% CI are presented in the Appendix A.

## 4. Discussion

To our knowledge, this is the first study assessing the relationships between breastfeeding and PWR in a nationally representative sample in Asia. Only 36.6% of women in our study returned to their pre-pregnancy weight after six months postpartum, however, this percentage was higher than has been previously reported—20% in Taiwan [21] and 13–20% in the US [1]. Different study populations and measurement methods could possibly explain the higher percentage of women returning to their pre-pregnancy weight in our study. Adverse birth outcomes for the mother and newborn were shown to prevent mothers from timely initiation of breastfeeding [25], which could influence their participation in breastfeeding surveys later; therefore an additional explanation could stem from the assumption that women in our study were healthier than the general population of mothers.

### 4.1. Duration and Exclusivity of Breastfeeding Are Beneficial for Weight Loss Postpartum

Our results support the hypothesis of dose–response relationships between breastfeeding and PWR. Each additional month of exclusive breastfeeding was associated with a reduction of 0.1–0.2 kg in PWR. Those who breastfed exclusively for 6 months retained 0.7 kg less weight than those who breastfed partially, and 1.3 kg less than those who did not breastfeed.

Previous studies reported significant associations between breastfeeding and PWR at 6–12 months postpartum of a magnitude comparable to our results [19,26,27,28], however the direct comparison was hampered by the diversity of breastfeeding assessments as only a few studies clearly separated their findings by the type of breastfeeding. Exclusive breastfeeding for at least 3 months resulted in 1.5 kg less PWR at 12 months postpartum, while partial breastfeeding had no significant effect in a national cohort of 2102 American women [27]. Each additional month of any type of breastfeeding contributed 0.44 kg to PWR reduction at 9 months postpartum in the Brazilian cohort (*N* = 405) [28], and each additional week of any breastfeeding contributed to 0.04 kg less weight retained at 12 months postpartum in the Australian cohort (*N* = 152) [19]. Several studies adapted lactation scores as a combined measurement of duration and exclusivity of breastfeeding. A significant correlation coefficient of 0.09 was observed between lactation intensity (sum of values assigned for each month: 0 if formula fed, 1 if mixed, and 2 if fully breastfed) and PWR in the Swedish cohort (*N* = 2295) [16]. Studies with similar methodology showed that each incremental point of lactation score contributed to a reduction of 0.03 kg in PWR at 12 months postpartum in the overweight and obese groups in the US [18], and to a reduction of 0.1 kg in PWR at 12 months postpartum in Brazil [26].

Two studies with large sample sizes (*N* > 5000) reported exclusive breastfeeding for six months to be associated with lower PWR at six months postpartum [29,30], while non-significant associations between breastfeeding and PWR were reported in studies with a smaller sample size (*N* < 500) [31,32]. Diversified study populations and methodology, high variability in breastfeeding definition, and postpartum timing of PWR assessment, as well as inadequate statistical power in some studies, might explain inconsistent results. In the future, it is recommended to uniformly define exclusive and partial breastfeeding to allow aggregation of knowledge and direct comparison. Since the effect of breastfeeding on PWR might be small, only studies with a large enough sample size could detect it, which should be kept mind during the design of future studies.

Since breastfeeding and postpartum weight loss occur simultaneously after delivery [15], establishing causal relationships is methodologically challenging; however, animal models and some human studies support hypotheses on the potential benefits of lactation on weight loss, explained by metabolic mechanisms and higher energy expenditure [33,34]. Findings from animal models and human studies suggest that lactation has a favorable effect on reducing PWR by reversing gestational metabolic changes more quickly and more completely [34]. While lactation may be beneficial to weight loss due to the mobilization of fat stores and an increase in energy expenditure, it also increases prolactin secretion, inducing a larger appetite and dietary intake [35]. Therefore, in developed countries where nutritious food is accessible for breastfeeding mothers, the influence of breastfeeding may not be easily observed in postpartum weight loss [36]. It is impossible to conduct a fully controlled randomized investigation of the topic due to ethical concerns, however some attempts have been done in Honduras, where exclusively breastfeeding women were randomized at four months postpartum to continue breastfeeding exclusively or to add nutritious solid foods. Those who exclusively breastfed for six months had significantly lower PWR than those who did so for four months [37].

Findings from diverse populations and methods support our results, which suggest a small but significant positive effect of breastfeeding duration and exclusivity on reducing PWR. Multiple benefits of exclusive breastfeeding have been well established [38] and exclusive breastfeeding for the first six months of life is being extensively recommended by the WHO [14]. Strict breastfeeding promotion of exclusive breastfeeding for six months have been recently criticized for inducing unnecessary stress on mothers, and an alternative idea of shifting the focus from an “all or nothing” approach towards an “every feed matters” one was suggested [39]. Our results support this more flexible approach to breastfeeding promotion by illustrating the additional benefits of postpartum weight control associated with each incremental time of breastfeeding, both exclusive and partial.

### 4.2. Entering Pregnancy with a Normal BMI Enhances the Effect of Breastfeeding on Weight Loss Postpartum

Negative dose–response relationships between exclusive breastfeeding and PWR were observed to a certain extent in all BMI groups; however, in women with a normal pre-pregnancy BMI, this relationship was most salient. Starting from one month postpartum, each additional month of exclusive breastfeeding contributed to a weight loss of 0.1–0.3 kg, and a similar relationship was observed with partial breastfeeding for ≥5 months. These estimates were lower than those obtained in a Danish population-based study on women with a normal BMI (18.5–25 kg/m^2^) [30], which may be accounted for by the ethnicity difference between Asian and western populations [40], and for the difference in the normal BMI definition in Taiwan (18.5–24 kg/m^2^). Entering pregnancy at a normal BMI may be associated with a healthier maternal behavior and environment as well as with more social support [13], which may confound the association between breastfeeding and PWR. Within this potential constraint, our results support the recommendation to enter pregnancy at a BMI within the normal range.

### 4.3. Obese Breastfeeding Women Retain Less Weight Postpartum

Obese women who exclusively breastfeed for six months retained 2.0 kg less postnatal weight compared to obese women who did not breastfed exclusively, while partial breastfeeding for six months showed only 0.2 kg less postnatal weight. This difference between exclusive and partial breastfeeding was the largest compared to other BMI groups. Exclusive breastfeeding for longer than 30 days and partial breastfeeding for longer than 180 days showed lower PWR compared to non-obese groups, respectively (Figure 3). Partial breastfeeding for 12 months in obese women was associated with weight loss of up to 2.1 kg, which was the largest compared to non-obese groups.

Our results are comparable to a Danish population-based study, where exclusively breastfeeding obese class I (30 ≤ BMI < 35) women reduced their weight by up to 2.4 kg at 6 months postpartum [30]. However, breastfeeding had little or no effect on postpartum weight retention in Brazilian obese women [28]. Controlling for multiple confounders including eating habits and physical activity nullified the association between breastfeeding and PWR at one year postpartum in American overweight and obese women [18]; however, another Danish study showed breastfeeding for 6 months to be associated with lower odds of excessive PWR (≥5 kg) in obese women, while adjusting for dietary intake and physical activity [41]. These inconsistent results could be explained by the diversity of study populations and methodologies.

Multiple studies, including the data of this study (Table 2), showed that women with an excessive pre-pregnancy weight are likely to gain less weight during pregnancy, and GWG is inversely associated with PWR [8,10,18]. This may partially explain our findings. Furthermore, some biological mechanism should be considered. For example, the higher metabolic rates in obese women accompanied by lactation [34] may explain the difference between obese and non-obese women in the association of breastfeeding and PWR. Another possible explanation may stem from the different patterns of fat weight change during the childbearing cycle and especially at the postpartum period in obese and non-obese women, e.g., skinfold thickness, waist-to-hip ratio and fat mass [42].

It is worth noting that obese women face multiple challenges regarding breastfeeding [43,44,45]. Pre-pregnancy obesity was associated with earlier breastfeeding cessation [43] and lower rates of breastfeeding initiation, duration, and exclusivity [44]. The proposed explanations included delayed lacto-genesis, insufficient supply of breast milk, low maternal self-efficacy to breastfeed [44], and lower prolactin response to suckling in the first week postpartum [45]. Therefore, we should not rule out the possible reverse causation, assuming that some obese women in our study, who successfully initiated breastfeeding and continued breastfeeding for some time postpartum, could have led a healthier lifestyle and enjoyed more psychosocial support contributing to their weight loss. Since many obese mothers may not initiate breastfeeding [44], they were less likely to participate in breastfeeding surveys. While the obese group in our study differed substantially from the non-obese groups, a longer duration of partial (>240 days) breastfeeding was required to observe significant changes in PWR within the group. We even observed a peak at 90 days of partial breastfeeding in PWR. This could be attributed to a smaller number of women in this group, resulting in wide and overlapping 95% CI in our models, implying no significant association between partial breastfeeding for up 240 days and PWR. We also assume that women with obesity were under-represented in our sample, which could have resulted in overestimated associations between breastfeeding and PWR in the obese group. This limitation can be addressed in future research focusing on women with excessive weight. Our results, in combination with previously known challenges to breastfeeding [43,44,45], suggest women with obesity as a potential target group for supportive interventions in breastfeeding promotion.

### 4.4. Study Limitations

Cross-sectional design of our study considerably limits causality inference and our conclusions must be viewed within the context of the additional limitations. Information bias was likely due to self-reported weight and height. Although self-reported weight measurements may be inaccurate on an individual level, self-reported pre-pregnancy weight of reproductive-aged women was found generally reliable and valid for population-based research [23]. A systematic review found that women tended to overestimate their height and underestimate their weight [46]. Therefore, the effect of breastfeeding on PWR could be overestimated given that the calculation of BMI was from self-reported weight and height in the study. However, if distorted self-reporting of weight and height was present before pregnancy and after birth, this would vary the bias. The self-reported duration of breastfeeding might be subject to recall bias, since our respondents were surveyed up to 14 months postpartum, and considerable time could have elapsed since they weaned their babies. Nevertheless, a large review of studies suggested that maternal recall is a valid and reliable estimate of breastfeeding duration within three years postpartum [47].

Previous studies in Taiwan reported that a combination of GWG with pre-pregnancy BMI and satisfaction with body image could explain 35% of the variance in PWR [48], and GWG, dietary intake, and long-term physical activity could explain 24% of PWR variation at 6 months and 27% of the variation at 12 months postpartum [49]. Although we adjusted our models for GWG, unmeasured confounding of lifestyles and dietary intake could attenuate the associations between breastfeeding and PWR. Since most of the respondents to breastfeeding surveys breastfed their babies, selection bias is also expected to underestimate our results.

## 5. Conclusions

Although there are well-established benefits of breastfeeding for mothers besides postpartum weight loss [38], our results provide additional support for breastfeeding duration and exclusivity. Following the notion of “every feed matters”, dose–response relationships may motivate mothers who are concerned with gaining extra weight after delivery to exclusively breastfeed longer. A longer duration of partial breastfeeding was shown to be beneficial for weight loss postpartum. Exclusive breastfeeding for up to 6 months and partial breastfeeding for a longer time may be recommended for obese mothers for better weight management. Future investigation into the effect of breastfeeding on PWR is recommended in diverse populations, employing longitudinal follow-up and standardized definitions of breastfeeding variables along with controlling for dietary intake and physical activity.

## Figures and Tables

**Figure 1 nutrients-12-01065-f001:**
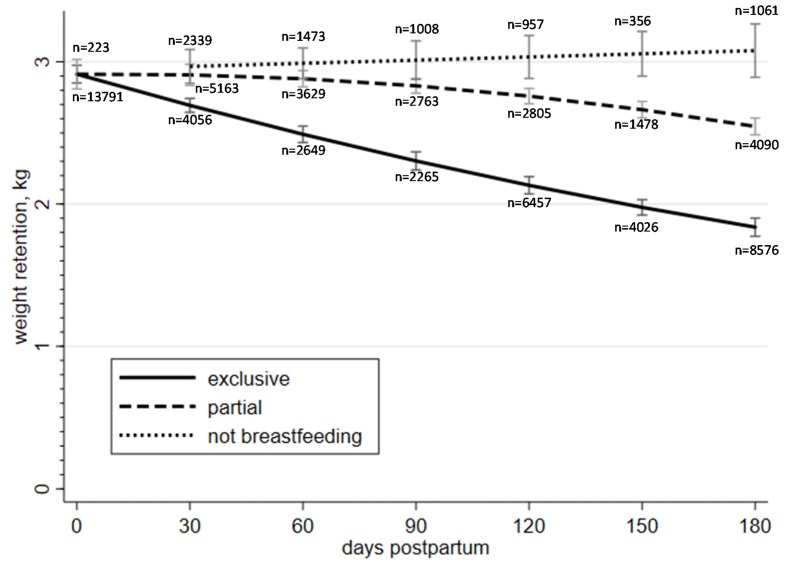
Adjusted means and 95% confidence intervals of postpartum weight retention and duration of breastfeeding in Taiwanese women (*N* = 52,367). Notes: (1) postpartum weight retention was assessed at 7–14 months postpartum; (2) X-axis represents duration of breastfeeding in days postpartum; (3) each time point on the same curve represents different women; (4) exclusively breastfeeding women that started adding food, liquids or formula are represented at a later time point on the curve for partial breastfeeding; (5) women that ceased breastfeeding completely are represented at a later time point on the curve for no breastfeeding; (6) sample sizes do not sum up to the total because of the cross-sectional design of the study.

**Figure 2 nutrients-12-01065-f002:**
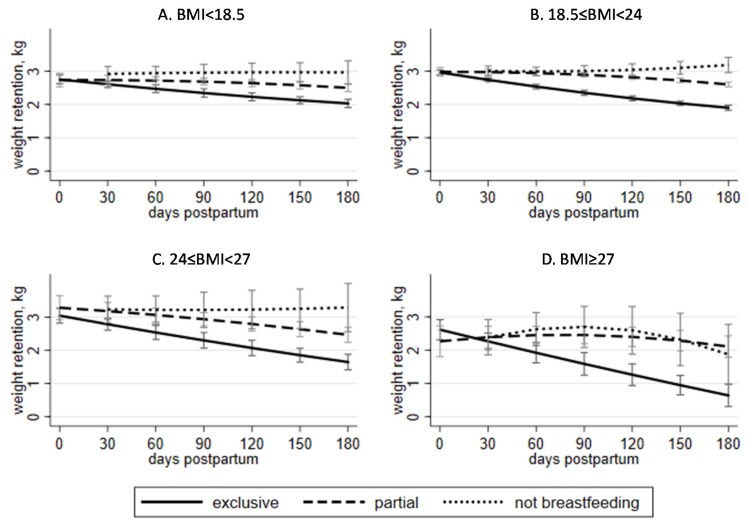
Adjusted means and 95% confidence intervals of postpartum weight retention and duration of breastfeeding in Taiwanese women by pre-pregnancy BMI (*N* = 52,367): (**A**). underweight (*n* = 8911), (**B**). normal (*n* = 34,927), (**C**). overweight (*n* = 5419), (**D**). obese (*n* = 3110). Notes: (1) postpartum weight retention was assessed at 7–14 months postpartum; (2) X-axis represents duration of breastfeeding in days postpartum; (3) each time point on the same curve represents different women; (4) exclusively breastfeeding women that started adding food, liquids, or formula are represented at a later time point on the curve for partial breastfeeding; (5) sample sizes do not sum up to the total because of the cross-sectional design of the study; (6) sample sizes for each time point are available in the Appendix A; (7) weight status categorized by BMI cutoffs for Taiwanese population—18.5, 24 and 27 kg/m^2^ (Taiwan Health Promotion Administration. (2018). BMI classification (in Chinese). Retrieved from http://health99.hpa.gov.tw/OnlinkHealth/Onlink_BMI.aspx).

**Figure 3 nutrients-12-01065-f003:**
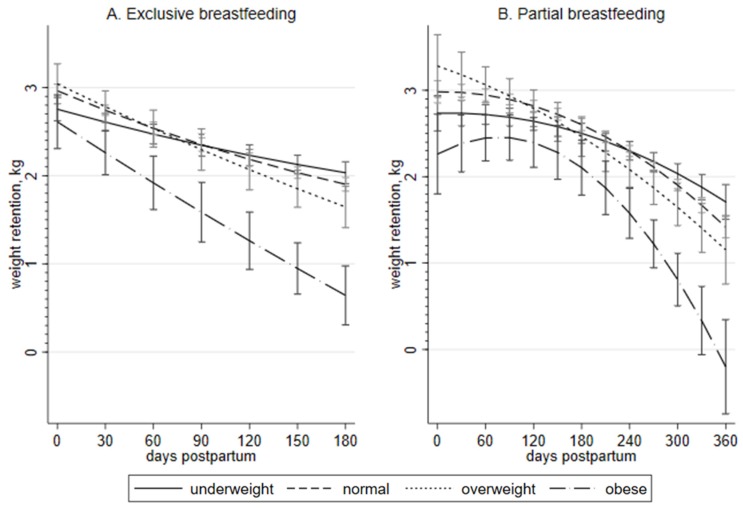
Postpartum weight retention and duration of breastfeeding in Taiwanese women by pre-pregnancy BMI (*N* = 52,367): (**A**). exclusive breastfeeding; (**B**). partial breastfeeding. Notes: (1) postpartum weight retention was assessed at 7–14 months postpartum; (2) X-axis represents duration of breastfeeding in days postpartum; (3) each time point on the same curve represents different women due to the cross-sectional design of the study; (4) sample sizes for each time point are available in the Appendix A; (5) weight status was categorized by BMI cutoffs for Taiwanese population—18.5, 24, and 27 kg/m^2^ (Taiwan Health Promotion Administration. (2018). BMI classification (in Chinese). Retrieved from http://health99.hpa.gov.tw/OnlinkHealth/Onlink_BMI.aspx).

**Table 1 nutrients-12-01065-t001:** Characteristics of breastfeeding survey respondents (*N* = 52,367).

	*n*	%	Mean PWR, kg	SD	*p*-Value *
Age					
20–24	3089	5.9	3.25	5.04	<0.001
25–29	12,245	23.4	2.82	4.42	
30–34	22,441	42.9	2.27	3.97	
35 or older	14,592	27.9	2.07	3.89	
Education					
Junior high or lower	1892	3.6	3.58	5.12	<0.001
High school	12,709	24.3	3.15	4.63	
Vocational school	8484	16.2	2.30	4.11	
University or higher	29,282	55.9	2.03	3.79	
Country of origin					
Taiwan	49,698	94.9	2.37	4.13	<0.001
Other	2669	5.1	3.02	4.30	
Employment					
Unemployed	22,066	42.1	2.55	4.45	<0.001
Employed	30,301	57.9	2.29	3.90	
Parity					
1	26,316	50.3	2.79	4.27	<0.001
>1	26,051	49.8	2.01	3.98	
Delivery					
Vaginal	33,905	64.7	2.31	4.03	<0.001
Caesarean	18,462	35.3	2.57	4.34	
Multiple gestation					
Yes	1373	2.6	2.42	4.15	<0.001
No	50,994	97.4	1.83	4.02	
Gestational weeks					
>37	48,124	91.9	2.41	4.13	0.016
≤37	4243	8.1	2.25	4.27	
Newborn sex					
Female	25,143	48.0	2.42	4.17	0.427
Male	27,224	52.0	2.39	4.13	
Low birth weight <2500 g					
Yes	3955	7.6	2.42	4.16	<0.001
No	48,412	92.5	2.17	3.97	
Was the newborn healthy at birth?					
Yes	44,082	84.2	2.38	4.14	0.050
No	8285	15.8	2.52	4.19	
Timing of the interview, months postpartum					
7	10,278	19.6	2.42	4.06	0.161
8	8638	16.5	2.35	4.04	
9	9533	18.2	2.32	4.13	
10	9011	17.2	2.36	4.19	
11	5164	9.9	2.48	4.20	
12	5760	11.0	2.28	4.17	
13	3163	6.0	2.46	4.45	
14	820	1.6	2.39	4.23	

BMI—body-mass index, PWR—postpartum weight retention, SD—standard deviation. * *p*-values are from t-tests for binary and from one-way analysis of variance for categorical variables.

**Table 2 nutrients-12-01065-t002:** Distribution of gestational weight gain, postpartum weight retention and breastfeeding duration by pre-pregnancy body mass index in Taiwanese women (*N*= 52,367).

	All	Underweight, BMI < 18.5	Normal, 18.5 ≤ BMI < 24	Overweight, 24 ≤ BMI < 27	Obese, BMI ≥ 27	
*n* (%)	52,367 (100.0)	8911 (17.0)	34,927 (66.7)	5419 (10.6)	3110 (5.9)	ANOVA *p*-value *
	Mean (SD)	Mean (SD)	Mean (SD)	Mean (SD)	Mean (SD)	
Gestational weight gain, kg	13.23 (5.05)	13.56 (4.81)	13.50 (4.92)	12.29 (5.40)	10.89 (5.65)	<0.001
Postpartum weight retention, kg	2.40 (4.15)	2.42 (3.40)	2.45 (4.01)	2.36 (5.13)	1.85 (5.43)	<0.001
Exclusive breastfeeding, days	82 (75)	80 (74)	84 (75)	81 (76)	74 (76)	<0.001
Partial breastfeeding, days	177 (112)	170 (113)	181 (111)	173 (113)	163 (114)	<0.001

* ANOVA—analysis of variance with Bonferroni pair-wise comparison, *p*-values indicate that means of the obese group are lower than any other group.

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
