# Peer review of "Dose-Response Relationships between Breastfeeding and Postpartum Weight Retention Differ by Pre-Pregnancy Body-Mass Index in Taiwanese Women"

_nutrients, 2020, doi:10.3390/nu12041065_

Round 1
Reviewer 1 Report
You stated on Page 2 line 64 -- "PWR is scare" then you list 12 Asian studies, that does not seem scarce to me.
Figure 2 -- it looks like there is a peak for 3 of the weight classes (where rate retention rises slightly peak at about 90 days then decreases, I think more discussion on this is warranted and intriguing really.
I am baffled (in a good way) that the Obese mothers lost the most weight - [I think you should have more focus in your discussion on this] but the Obese group typically gain the least weight during pregnancy so, if true this would mean they actually lost pre-pregnancy weight v.s just lost what they gained during pregnancy - which I think it quite intriguing.
There are some details that are unclear to me:
- For the PWR at what time point was it cut i.e. PWR as defined as pre-pregnancy weight and weight at 6-mos post or 1-mo post or when was it (it is a bit unclear to me)?
- I think ALL non-singleton births should be removed from analyses.
- I would like a table that has the breakdown of PWR by month (is that what the Supplamental files are (they are not labeled thus, unclear) - if yes (and I think it may be) I think you should go further than simply the tenths place but the hundreth
Author Response
Dear reviewer, we are very grateful for your time dedicated to reading our manuscript. Your comments provided multiple insights and helped us to improve our writing. We tried our best to carefully address your comments. Kindly refer to the details below.
You stated on Page 2 line 64 -- "PWR is scarce" then you list 12 Asian studies that does not seem scarce to me.
-We rephrased as: Some studies in Asian countries, including Taiwan, focused on PWR, however only a few examined its relationship with breastfeeding, lines 67-68
Figure 2 -- it looks like there is a peak for 3 of the weight classes (where rate retention rises slightly peak at about 90 days then decreases, I think more discussion on this is warranted and intriguing really.
-A peak at 90 days was observed for partial breastfeeding in the obese group (Figures 2D, 3B). Since 95% CI overlap at the breastfeeding duration up to 240 days postpartum, we interpret this result as no significant change in PWR associated with this duration of partial breastfeeding. We noted this in the results, lines 204-205, 238-239 and added to the discussion, lines 374-376
I am baffled (in a good way) that the Obese mothers lost the most weight - [I think you should have more focus in your discussion on this] but the Obese group typically gain the least weight during pregnancy so, if true this would mean they actually lost pre-pregnancy weight v.s just lost what they gained during pregnancy - which I think it quite intriguing.
-We added a paragraph to the discussion lines 354-362:
Multiple studies, including the data of this study (Table 2) showed that women with excessive pre-pregnancy weight are likely to gain less weight during pregnancy, and GWG is inversely associated with PWR [8, 10, 18]. This may partially explain our findings. Furthermore, some biological mechanism should be considered. For example, the higher metabolic rates in obese women accompanied by lactation [35] may explain the difference between obese and non-obese women in association of breastfeeding and PWR. Another possible explanation may stem from the different patterns of fat weight change during the childbearing cycle and especially at the postpartum period in obese and non-obese, e.g., skinfold thickness, waist-to-hip ratio and fat mass [45].
There are some details that are unclear to me:
1. For the PWR at what time point was it cut i.e. PWR as defined as pre-pregnancy weight and weight at 6-mos post or 1-mo post or when was it (it is a bit unclear to me)?
-We clarified that PWR was calculated by subtracting pre-pregnancy weight from the weight reported at time of the interview. Since interviews were held at different times postpartum, ranging from 7 to 14 months, we compared PWR means for each month with analysis of variance (ANOVA), resulting in a non-significant difference (p = 0.161) (Table 1), Methods, lines 110-112
2. I think ALL non-singleton births should be removed from analyses.
-Having twins or multiple gestations can influence the weight gained during pregnancy and affect breastfeeding behavior. Since we considered PWR as our outcome of interest, we expected the effect of twins to be negligible, especially given their relatively small proportion (2.6%) in our dataset. The adjustment for multiple gestations and gestational weight gain provided very similar results to the analysis without non-singleton births. Since our conclusions remained unaffected, we kept the results based on the full dataset. We added explanation to the lines 145-149 in section “2.3. Data analysis”, and provided the original results with non-singletons births excluded in the supplementary materials (Figures S4-S6).
3. I would like a table that has the breakdown of PWR by month (is that what the Supplemental files are (they are not labeled thus, unclear) - if yes (and I think it may be) I think you should go further than simply the tenths place but the hundredth
-Breakdown of PWR by month postpartum (at the time of the interview) is presented in Table 1 and by breastfeeding duration (in intervals of 30 days) in supplementary tables (S1-S3). We labeled the supplementary tables accordingly and added the hundredth digit to the mean PWR throughout all the manuscript.
Reviewer 2 Report
Nutrients March 2020
Dose-response relationships between breastfeeding 2 and postpartum weight retention differ by pre-3 pregnancy body-mass index in Taiwanese women 4
Alexander Waits1, Chao-Yu Guo1, Yan-Shing Chang2, Li-Yin Chien1, 3*
Overall
This paper is very interesting, and on an important topic, as the relationship between PWR and breastfeeding has been of some debate.
Minor issues with English language throughout.
Abstract
Suggest ‘fully understood’ rather than ‘confirmed’.
Line 25: this sentence is confusing: “For those who breastfed exclusively for >30 days or partially for >180 days, obese women had 25 a lower PWR than any other BMI group.”
I wonder if it would be better to present results that are easier to understand and have more impact, such as how much difference in PWG over the 6 months for each of the BMI categories.
The results in this paper were really interesting, but I don’t think this is conveyed well in the abstract. Also, the way the results are presented feels like they are being cherry-picked.
Intro
P 1 line 35: Sentence makes no sense: “ do not return to their pre-pregnancy weight or even keep excessive weight”
Line 58, I find this type of language unclear: “significant negative associations between breastfeeding and PWR”
Introduction could be written with greater clarity.
Methods
P2 line 93: please clarify meaning of disproportionate. Were some groups over-sampled? If so, please state which and why: “Disproportionate probability sampling method was applied, based on annual number of births in 25 Taiwan counties”
Line 97: given that women were sampled at 7-14 months postpartum, this question does not make sense if the person was still breastfeeding. “1) “Till what time (days postpartum) did you breastfeed?”” How did you treat data where the woman was still breastfeeding?
Pre-pregnancy weight was retrospective. Please provide a reference for why this is okay in the methods (I’ve seen it in the limitations section but needs to be explained here.)
P 3 line 128, why was 30 days chosen? “We compared PWR means for every 30 days of breastfeeding to assess statistical significance.” Was it not possible to use continuous data?
Results
Table 1- the 4th column states Mean PWR in kg, however this may be correct for the lower results (from age onwards) but not for the first section. Please revise with suitable headings.
Figure is good- clearly shows the number of persons contacted at each time point and the differences between groups plus the 95% CIs. Suggest you make the legend smaller, in line with the font size of the rest of the graph. Perhaps ‘no breastfeeding’ should be ‘not breastfeeding’, given that persons can move between groups. I’m assuming that there were differences in the amount of women being contacted at the different time-points? If so, please be explicit about this.
3.3 stratification by PP BMI. Why is the ‘not breastfeeding’ group not represented in the figures at Figure 2?
Figure 3- why is partial breastfeeding but not exclusive breastfeeding presented in this format?
Discussion
P 8/9 line 235 “Adverse birth outcomes for the mother and newborn were shown to prevent mothers from timely initiation of breastfeeding [26] which could influence their participation in breastfeeding surveys later; therefore, we assumed that women in our study were healthier than the general population of mothers.”
I don’t think this conclusion is necessarily correct. The samples may be different on a number of factors, and there could have been other issues with previous data. There could be a number of reasons for this.
Paragraphs at 4.1 needs refining. The information is stacked on paper after paper- it is hard to read and make sense of it. Paragraph structure poor, and needs to be rewritten.
P 9 line 269: “Since breastfeeding and weight loss occur simultaneously,” Do you mean in the current sample- (if so please clarify), or known generally- (if so please reference)?
Page 9 line 276- excellent point: “Therefore, in developed countries, where nutritious 276 food is accessible for breastfeeding mothers, the influence of breastfeeding may not be easily 277 observed in postpartum weight loss [37].”
Line 279. Both these are correct, but unrelated and should not be in the same sentence: “Randomized trials are nearly non-existent due to ethical concerns, yet low-income primiparous 279 women in Honduras, who exclusively breastfed for 6 months, had significantly lower PWR than 280 those who did so for 4 months [38].”
Line 283 Also an excellent point, but is buried in the middle of an unrelated paragraph: “Strict breastfeeding promotions have been recently criticized for inducing 283 unnecessary stress on mothers, and an alternative idea of shifting the focus from “all or nothing” 284 towards “every feed matters” approach was suggested [39].”
Line 297: one sentence is not a paragraph. Join with above: “Entering pregnancy at a normal BMI may be associated with a healthier maternal behavior and 297 environment as well as with more social support [13], which may confound the association between 298 breastfeeding and PWR.”
P 10 Line 316. Try two sentences here, and use commas not semi-colons: “Pre-pregnancy obesity was 316 associated with earlier breastfeeding cessation [42] and lower rates of breastfeeding initiation, 317 duration and exclusivity, explained by delayed lacto-genesis; insufficient supply of breast milk; low 318 maternal self-efficacy to breastfeed [43]; and lower prolactin response to suckling in the first week 319 postpartum [44].”
Line 323 should be ‘to participate’: “Since many obese mothers did not initiate breastfeeding [42, 43], they were likely to refuse 323 participating in breastfeeding surveys.”
Line 325:” Although this could result in overestimated associations between breastfeeding and 325 PWR, supporting breastfeeding in obese women is still warranted for the additional benefit of 326 potential weight reduction.” These are two separate concepts and should not be in the same sentence.
Line 325: needs a subheading “limitations” at: “Cross-sectional design…”
Conclusion
Please mark the conclusion as such with a subheading.
Conclusion is good, but should be all one paragraph. Do not have one sentence paragraphs.
Author Response
Dear reviewer, we are very grateful for your time dedicated to reading our manuscript. Your comments provided multiple insights and helped us to improve our writing. We tried our best to carefully address your comments. Kindly refer to the details below.
Overall
This paper is very interesting, and on an important topic, as the relationship between PWR and breastfeeding has been of some debate.
Minor issues with English language throughout.
-We asked a native English speaker to check the manuscript once more before the resubmission.
Abstract
Suggest ‘fully understood’ rather than ‘confirmed’.
-We replaced according to the reviewer’s suggestion, line 14
Line 25: this sentence is confusing: “For those who breastfed exclusively for >30 days or partially for >180 days, obese women had 25 a lower PWR than any other BMI group.”
-We replaced: Women with obesity, who breastfed exclusively for >30 or partially for >180 days, had lower PWR than non-obese groups
I wonder if it would be better to present results that are easier to understand and have more impact, such as how much difference in PWG over the 6 months for each of the BMI categories.
The results in this paper were really interesting, but I don’t think this is conveyed well in the abstract. Also, the way the results are presented feels like they are being cherry-picked.
-We tried to follow the reviewer’s comments regarding the abstract, while still staying within the word count limit of 200 as instructed by the authors’ guidelines. We have rewritten the results presented in the abstract following our study objectives to avoid ‘cherry picking’, lines 20-28.
Intro
P 1 line 35: Sentence makes no sense: “do not return to their pre-pregnancy weight or even keep excessive weight”
- We rephrased as: “Many postpartum women keep some of the weight gained during gestation, which prevents them from returning to their pre-pregnancy weight”, lines 36-37
Line 58, I find this type of language unclear: “significant negative associations between breastfeeding and PWR”
-We rephrased the sentence as “significant negative associations between breastfeeding duration/frequency and PWR, lines 61-62
Introduction could be written with greater clarity.
-We rewrote the introduction, especially paragraphs 1, 2 and 4 to improve the clarity
Methods
P2 line 93: please clarify meaning of disproportionate. Were some groups over-sampled? If so, please state which and why: “Disproportionate probability sampling method was applied, based on annual number of births in 25 Taiwan counties”
- We changed to: Disproportionate probability sampling method, based on annual number of births in 25 Taiwan counties, was applied so that county-specific breastfeeding rate could be estimated for counties with small number of births, line 95
Line 97: given that women were sampled at 7-14 months postpartum, this question does not make sense if the person was still breastfeeding. “1) “Till what time (days postpartum) did you breastfeed?”” How did you treat data where the woman was still breastfeeding?
-We added: If the answer was “I am still breastfeeding”, breastfeeding duration was taken as the days postpartum at the time of the interview, lines 101-103
Pre-pregnancy weight was retrospective. Please provide a reference for why this is okay in the methods (I’ve seen it in the limitations section but needs to be explained here.)
-We provided the reference in the methods and added: Pre-pregnancy BMI in kg/m² was calculated from self-reported pre-pregnancy weight and height, which was found as a reliable and valid method for population-based research in women of reproductive age. Lines 113-115
Shin, D.; Chung, H.; Weatherspoon, L.; Song, W.O. Validity of pre-pregnancy weight status estimated from self-reported height and weight. Maternal and child health journal 2014, 18, 1667-1674.
P 3 line 128, why was 30 days chosen? “We compared PWR means for every 30 days of breastfeeding to assess statistical significance.” Was it not possible to use continuous data?
-We used continuous data to fit our models, however, because of non-linear relationships, assessment and comparison of slopes could be applied only at specific points. We therefore choose 30 days or one month as a reasonable time interval to obtain estimates of relationships between breastfeeding and PWR. We added this clarification to the methods, lines 133-134.
Results
Table 1- the 4th column states Mean PWR in kg, however this may be correct for the lower results (from age onwards) but not for the first section. Please revise with suitable headings.
-We split table 1 to table 1 and table 2 to provide more detailed description of our study population stratified by BMI.
Figure is good- clearly shows the number of persons contacted at each time point and the differences between groups plus the 95% CIs. Suggest you make the legend smaller, in line with the font size of the rest of the graph. Perhaps ‘no breastfeeding’ should be ‘not breastfeeding’, given that persons can move between groups. I’m assuming that there were differences in the amount of women being contacted at the different time-points? If so, please be explicit about this.
-We made legend the same font size as the rest of the graph, replaced ‘no breastfeeding’ with ‘not breastfeeding’, and clarified in all the 3 figures that women in our study were contacted at 7-14 months postpartum and X-axis represents breastfeeding duration. The differences in the amount of women being contacted at the different time-points are presented in table 1.
3.3 stratification by PP BMI. Why is the ‘not breastfeeding’ group not represented in the figures at Figure 2?
-We added the ‘not breastfeeding’ group to figure 2, and the corresponding details in results and supplementary files (Table S2, Figure S2).
Figure 3- why is partial breastfeeding but not exclusive breastfeeding presented in this format?
-We extended figure 3 to figure 3A with exclusive breastfeeding and figure 3B with partial breastfeeding, and added the corresponding details in methods, results and discussion.
Discussion
P 8/9 line 235 “Adverse birth outcomes for the mother and newborn were shown to prevent mothers from timely initiation of breastfeeding [26] which could influence their participation in breastfeeding surveys later; therefore, we assumed that women in our study were healthier than the general population of mothers.”
I don’t think this conclusion is necessarily correct. The samples may be different on a number of factors, and there could have been other issues with previous data. There could be a number of reasons for this.
-We added other potential reasons for this, line 263: Different study populations and measurement methods could possibly explain the higher percentage of women returning to their pre-pregnancy weight in our study.
Paragraphs at 4.1 needs refining. The information is stacked on paper after paper- it is hard to read and make sense of it. Paragraph structure poor, and needs to be rewritten.
-We have rewritten paragraph 4.1 based on the reviewer’s comments above and below
P 9 line 269: “Since breastfeeding and weight loss occur simultaneously,” Do you mean in the current sample- (if so please clarify), or known generally- (if so please reference)?
-We specified this is a general fact and provided a reference in line 300
Page 9 line 276- excellent point: “Therefore, in developed countries, where nutritious 276 food is accessible for breastfeeding mothers, the influence of breastfeeding may not be easily 277 observed in postpartum weight loss [37].”
Line 279. Both these are correct, but unrelated and should not be in the same sentence: “Randomized trials are nearly non-existent due to ethical concerns, yet low-income primiparous 279 women in Honduras, who exclusively breastfed for 6 months, had significantly lower PWR than 280 those who did so for 4 months [38].”
-We separated these statements and rephrased the statement in lines 309-314
Line 283 Also an excellent point, but is buried in the middle of an unrelated paragraph: “Strict breastfeeding promotions have been recently criticized for inducing 283 unnecessary stress on mothers, and an alternative idea of shifting the focus from “all or nothing” 284 towards “every feed matters” approach was suggested [39].”
-We changed to:
Strict breastfeeding promotion of exclusive breastfeeding for six months have been recently criticized for inducing unnecessary stress on mothers, and an alternative idea of shifting the focus from “all or nothing” towards “every feed matters” approach was suggested [39]. Our results support this more flexible approach to breastfeeding promotion by illustrating the additional benefit of postpartum weight control associated with each incremental time of breastfeeding, both exclusive and partial. Lines 318-323
Line 297: one sentence is not a paragraph. Join with above: “Entering pregnancy at a normal BMI may be associated with a healthier maternal behavior and 297 environment as well as with more social support [13], which may confound the association between 298 breastfeeding and PWR.”
-We joined the sentences into one paragraph
P 10 Line 316. Try two sentences here, and use commas not semi-colons: “Pre-pregnancy obesity was 316 associated with earlier breastfeeding cessation [42] and lower rates of breastfeeding initiation, 317 duration and exclusivity, explained by delayed lacto-genesis; insufficient supply of breast milk; low 318 maternal self-efficacy to breastfeed [43]; and lower prolactin response to suckling in the first week 319 postpartum [44].”
-We separated the sentence according to the reviewer’s comment
Line 323 should be ‘to participate’: “Since many obese mothers did not initiate breastfeeding [42, 43], they were likely to refuse 323 participating in breastfeeding surveys.”
-We changed according to reviewer’s comment
Line 325:” Although this could result in overestimated associations between breastfeeding and 325 PWR, supporting breastfeeding in obese women is still warranted for the additional benefit of 326 potential weight reduction.” These are two separate concepts and should not be in the same sentence.
-We separated the sentence according to reviewer’s comment
Line 325: needs a subheading “limitations” at: “Cross-sectional design…”
-We added the subheading “Study limitations”
Conclusion
Please mark the conclusion as such with a subheading.
Conclusion is good, but should be all one paragraph. Do not have one sentence paragraphs.
-We added the subheading and made the section one paragraph.